 eLIFE

# Enhanced homology-directed human genome engineering by controlled timing of CRISPR/Cas9 delivery

**Steven Lin[1†], Brett T Staahl[1†], Ravi K Alla[2], Jennifer A Doudna[1,3,4,5]\***

[1]Department of Molecular and Cell Biology, University of California, Berkeley, Berkeley, United States; [2]Computational Genomics Resource Laboratory, QB3, University of California, Berkeley, Berkeley, United States; [3]Howard Hughes Medical Institute, University of California, Berkeley, Berkeley, United States; [4]Department of Chemistry, University of California, Berkeley, Berkeley, United States; [5]Department of Chemistry, Lawrence Berkeley National Laboratory, Berkeley, United States

**Abstract** The CRISPR/Cas9 system is a robust genome editing technology that works in human cells, animals and plants based on the RNA-programmed DNA cleaving activity of the Cas9 enzyme. Building on previous work (*Jinek et al., 2013*), we show here that new genetic information can be introduced site-specifically and with high efficiency by homology-directed repair (HDR) of Cas9-induced site-specific double-strand DNA breaks using timed delivery of Cas9-guide RNA ribonucleoprotein (RNP) complexes. Cas9 RNP-mediated HDR in HEK293T, human primary neonatal fibroblast and human embryonic stem cells was increased dramatically relative to experiments in unsynchronized cells, with rates of HDR up to 38% observed in HEK293T cells. Sequencing of on- and potential off-target sites showed that editing occurred with high fidelity, while cell mortality was minimized. This approach provides a simple and highly effective strategy for enhancing site-specific genome engineering in both transformed and primary human cells.

**\*For correspondence:** doudna@berkeley.edu

[†]These authors contributed equally to this work

**Competing interests:** The authors declare that no competing interests exist.

**Reviewing editor**: Detlef Weigel, Max Planck Institute for Developmental Biology, Germany

## Introduction

The CRISPR-associated enzyme Cas9 enables site-specific genome engineering by introducing double-strand breaks (DSB) at guide RNA-specified chromosomal loci of interest (*Cong et al., 2013*; *Jinek et al., 2013*; *Mali et al., 2013a*). Cells repair DSBs using the non-homologous end joining (NHEJ) or homology-directed repair (HDR) pathways. The NHEJ pathway generates variable insertions or deletions (indels) at the DSB, while HDR employs homologous donor DNA sequences from sister chromatids, homologous chromosomes or exogenous DNA molecules to produce precise insertions, deletions or base substitutions at a DSB site or between two DSBs. Such precise modifications are desired for targeted genome engineering.

Although cells have differing abilities to repair DSBs using NHEJ or HDR, the phase of the cell cycle largely governs the choice of pathway. NHEJ dominates DNA repair during G1, S and G2 phases, whereas HDR is restricted to late S and G2 phases when DNA replication is completed and sister chromatids are available to serve as repair templates (*Heyer et al., 2010*). Impediments to HDR include competition with NHEJ in S and G2 phases and specific down-regulation of HDR at M phase and early G1 to prevent telomere fusion (*Orthwein et al., 2014*). Although chemical or genetic interruption of the NHEJ pathway can favor HDR (*Shrivastav et al., 2008*), such manipulations can be difficult to employ, harmful to cells or both. Consequently, high cleavage activity of programmable nucleases does not necessarily correlate with efficient HDR-induced genome editing.

Here we report a simple and robust approach that advances our previous findings (*Jinek et al., 2013*) to enhance HDR efficiency in human cells. This strategy combines well-established cell cycle synchronization techniques with direct nucleofection of pre-assembled Cas9 ribonucleoprotein (RNP) complexes to achieve controlled nuclease action at the phase of the cell cycle best for HDR (*Figure 1A*). HEK293T, human primary neonatal fibroblast and H9 human embryonic stem cells demonstrated robust HDR-mediated genome editing at levels up to 38% with no detected off-target editing. These results establish a superior approach to Cas9-mediated human genome engineering that enables efficient mutation, repair and tagging of endogenous loci in a rapid and predictable manner.

## Results

To test whether S phase is optimal for HDR in HEK293T cells, six reversible chemical inhibitors were used in parallel experiments to synchronize HEK293T cells at G1, S and M phases of the cell cycle, followed by release prior to nucleofection with Cas9 RNP (*Figure 1B*, *Figure 1—figure supplement 1A*). Immediately after release we prepared 30-µl nucleofection reactions containing $2 \times 10^5$ cells, Cas9 RNP with sgRNA targeting EMX1 gene and a 183-nucleotide single-stranded oligonucleotide DNA (ssODNA) HDR template (*Figure 1C*). After 24 hr, cells were analyzed for HDR (specifically, exogenous donor template mediated HDR) or total editing (TE, defined as the sum of all NHEJ and HDR events that give rise to indels) at the Cas9 cleavage site within EMX1, showing that both aphidicolin and nocodazole led to pronounced increases in Cas9-mediated editing frequencies (*Figure 1D,E*). The enhancement is more evident at lower Cas9 RNP concentration (30 pmol), improving HDR rates from ~9% in unsynchronized cells to ~14% with aphidicolin and ~20% with nocodazole (*Figure 1E*). The highest HDR frequency achieved was 31% with nocodazole synchronization and 100 pmol of Cas9 RNP. Importantly, 1 day after nocodazole release the synchronized cells were cycling like unsynchronized controls and appeared morphologically normal (*Figure 1—figure supplement 1A*).

Next we determined systematically the dosage effect of Cas9 RNPs and HDR templates on HDR efficiency in control and nocodazole synchronized cells. At the EMX1 locus, we tested three concentrations of Cas9 RNP (10, 30 and 100 pmol) in combination with three concentrations of HDR template (50, 100 and 200 pmol in *Figure 1C*). The overall frequencies of TE and HDR increased proportionally with increasing Cas9 RNP concentration (*Figure 2A*). Synchronization increased the TE frequency twofold at 10 pmol and 1.5-fold at 30 pmol Cas9 RNP, but the enhancement diminished at 100 pmol. The HDR frequency also increased dramatically with synchronization, especially at lower concentrations of Cas9 RNP, from undetectable to 9–15% at 10 pmol of Cas9 RNP, and from 6–12% to 22–28% at 30 pmol (*Figure 2A*). These results demonstrate that timed delivery of Cas9 RNP into M-phase synchronized HEK293T cells enhances HDR by several fold above the levels observed without synchronization.

To test these effects at other genomic loci, we programmed Cas9 RNPs to target the DYRK1 gene, which is important for brain development, autism and Downs Syndrome (*Arron et al., 2006*; *Fotaki et al., 2002*; *O'Roak et al., 2012*). We assayed two ssODNA HDR templates spanning the same sequence but with different orientations: one identical to the target strand sequence (+strand) and the other its complement (−strand) (*Figure 2B*). Both of these templates yielded comparable levels of HDR. Strikingly, nocodazole synchronization enhanced the TE frequencies more than twofold and HDR frequencies over sixfold at all doses of Cas9 RNP (*Figure 2B*). Moreover, nocodazole synchronization reduced the requirement for high Cas9 RNP concentrations, producing 5–6% HDR at 10 pmol of Cas9; 10-fold more Cas9 RNP was required to achieve the same HDR frequency in unsynchronized cells.

We also programmed Cas9 RNPs to target the CXCR4 gene, a chemokine receptor implicated in HIV entry (*Feng et al., 1996*) and cancer metastasis (*Murphy, 2001*). The HDR template used in these experiments was a ssODNA oriented complementary (−strand) to the target strand, containing HindIII and BamHI restriction sites flanked by 90 nt homology arms. The enhancement in TE and HDR frequencies at CXCR4 was comparable to those observed for the DYRK1 target site (*Figure 2C*). The most significant increase was again observed at 10 and 30 pmol of Cas9 RNP, yielding nearly five and twofold increases, respectively. In this case, nocodazole synchronization yielded 27% HDR at 10 pmol of Cas9 RNP. A comparable level of HDR in the unsynchronized cells would require 100 pmol of RNP. Collectively, the results from EMX1, DYRK1 and CXCR4 loci demonstrate that nocodazole synchronization is a highly effective and broadly applicable method to enhance the TE and HDR frequencies in HEK293T cells.

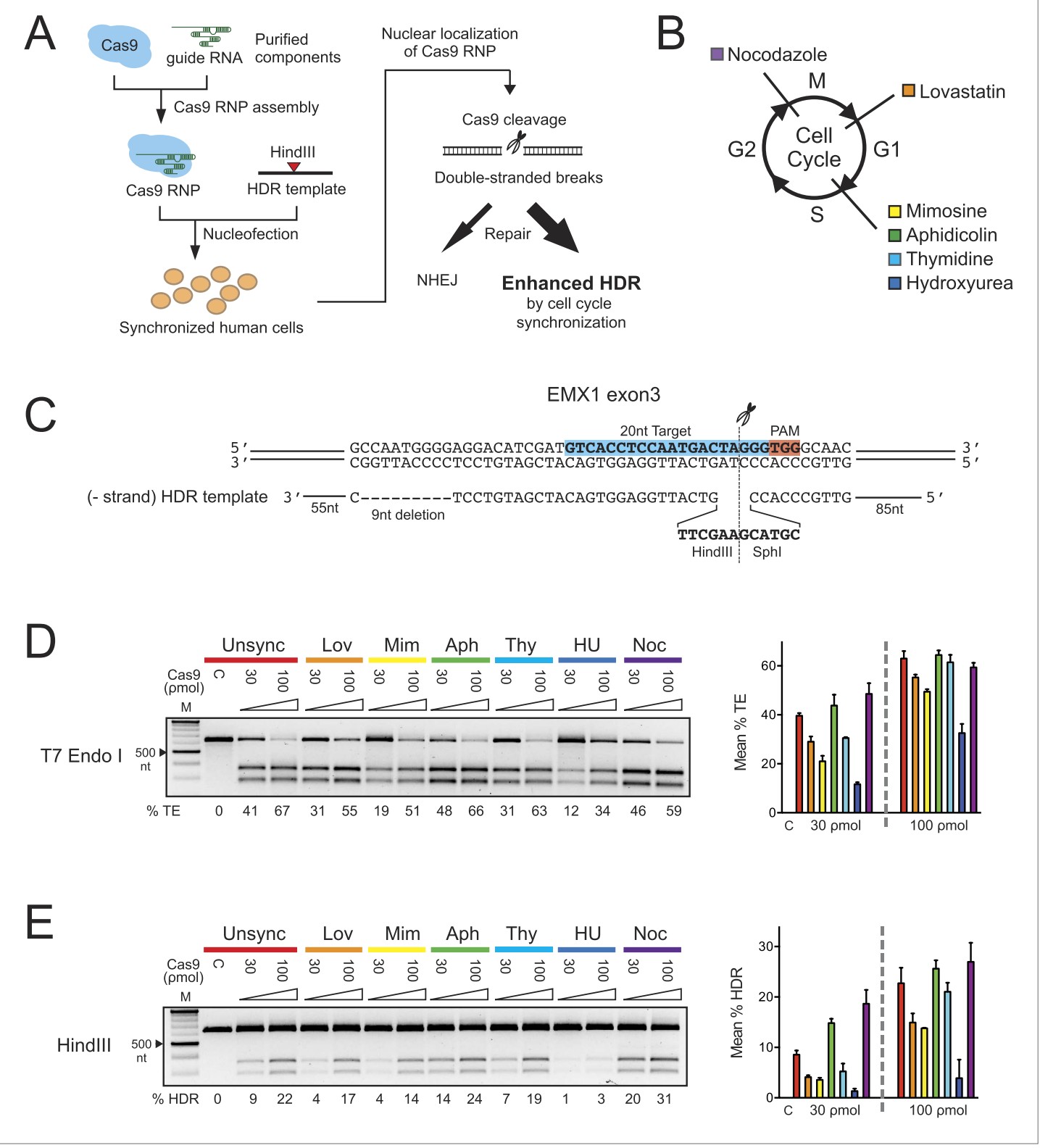

**Figure 1**. The effect of cell cycle synchronization on total editing and homology-directed repair frequencies in HEK293T cells. (**A**) Experimental schematic of timed delivery of Cas9-guide RNA ribonucleoprotein (RNP) into human cells for genome editing. (**B**) Chemical inhibitors used to arrest cells at specific phases of cell cycle included lovastatin (Lov), which blocks at early G1 and partially at G2/M phase; mimosine (Mim), aphidicolin (Aph), thymidine (Thy) and hydroxyurea (HU) which arrest cells at the G1-S border prior to onset of DNA replication; and nocodazole (Noc) which causes arrest
*Figure 1. Continued on next page*

*Figure 1. Continued*

at G2/M phase. (**C**) The homology-directed repair (HDR) donor DNA is a 183 nt ssODNA that is complementary to the target sequence (−strand) and contains a 9 nt insertion (HindIII and SphI restriction sequences) at the cut site and a 9 nt deletion downstream of the cut site; these modifications are flanked by 85 nt and 55 nt asymmetrical homology arms at 5′ and 3′ ends, respectively. (**D**, **E**) PCR-based screening of cell cycle inhibitors for enhancement of Cas9-triggered total editing (TE) (**D**) and HDR (**E**) frequencies in HEK293T cells. For each inhibitor condition (color coded), two doses of Cas9 RNP, 30 and 100 ρmol, were transfected with 100 ρmol of HDR DNA template; control reactions (labeled as C) contained 100 ρmol of Cas9 but no sgRNA. The TE frequency was measured using a T7 endonuclease I assay and analyzed using a formula described in 'Materials and Methods'. The HDR frequency was determined directly by HindIII digestion, which specifically cleaved the newly integrated HindIII sequence, and calculated as the ratio of DNA product to DNA substrate. The % TE, % HDR and standard deviation (error bars) were calculated from three experiments.

The following figure supplement is available for figure 1:

**Figure supplement 1**. FACS analysis reveals cell cycle blocks and the DNA content in the cells that are arrested at different phases of cell cycle.

Off-target editing increases with increasing nucleic acid-based delivery of Cas9 (***Fu et al., 2013***; ***Hsu et al., 2013***; ***Pattanayak et al., 2013***; ***Mali et al., 2013b***). We reasoned that the short-lived Cas9 RNP (***Kim et al., 2014***) and timed delivery would minimize off-target editing and that potential toxicity might be minimized by using lower RNP amounts. We compared the TE and HDR frequencies at the EMX1 and DYRK1 target loci to those occurring at the top two predicted off-target loci (***Hsu et al., 2013***) respectively by deep sequencing a representative biological replicate experiment from ***Figure 2A,B***. Importantly, no off-target editing was detected above background levels under all conditions, and increasing RNP dosage had no effect on off-target editing. As shown in ***Figure 2—figure supplement 1A***, ***Supplementary file 1***, the TE (indels/total reads) and HDR (HDR/total reads) frequencies at EMX1 and DYRK1 increased with RNP dosage in both cell conditions. Most importantly, there was no detectable HDR at the off-target loci. Overall, the TE and HDR frequencies detected by deep sequencing were comparable to our previous gel densitometry results (***Figure 2A,B***). The deep sequencing analysis also allowed us to determine the ratio of Cas9-induced DSBs being repaired by the NHEJ vs HDR pathway (HDR/TE reads). Although nocodazole synchronization increased the HDR/TE ratio by twofold at the EMX1 locus and fivefold at the DYRK1 locus, this ratio reached a maximum value of ~33% across all RNP doses, suggesting that the maximum capacity of the HDR machinery in HEK293T cells is to repair ~33% of DSBs. A panel of representative indels is shown in ***Figure 2—figure supplement 1B***.

To examine the length of HDR template homology sequences required for Cas9-mediated HDR, we tested four single-stranded and two double-stranded HDR templates for EMX1 bearing homology arms ranging from 30 to 250 nt in length (template 2–7 in ***Figure 3A***). To avoid signal saturation and better distinguish the HDR frequencies of different templates, we reduced the Cas9 RNP and HDR template concentrations to 30 ρmol and 50 ρmol, respectively. As observed previously, nocodazole synchronization produced higher HDR frequencies than observed in the unsynchronized cells (***Figure 3B***). In addition to the unsynchronized and nocodazole synchronized conditions, we included a third condition in which aphidicolin, an S-phase blocker, was added to the nocodazole synchronized cells immediately after nucleofection. We hypothesized that the aphidicolin-blocked cells would show reduced HDR frequency due to the inability to enter S phase where HDR is thought to be most active. As expected, the aphidicolin block significantly reduced the HDR frequency, supporting the conclusion that cells need to proceed through S phase, and possibly G2 as well, for highly efficient HDR.

DNA molecules with at least 60 nt of sequence homology flanking the Cas9 cleavage site were sufficient for highly efficient HDR of ~19% (***Figure 3B***). Further extension of the homology arms to 90 nt increased the HDR frequency only slightly (~20–23%). Both (+) and (−) template orientations were similarly effective, as also observed in the DYRK1 experiment. When double-stranded templates 6 and 7 were used, HDR frequencies were reduced to 7%. Moreover, unusual banding patterns in the HindIII HDR assay (***Figure 3B***) implied the presence of a concatemerized HDR template or non-specific recombination products, at least with short sequences as employed here.

To expand our findings on the cell cycle synchronization method to other cell types, we targeted the EMX1 gene in human primary neonatal fibroblasts (neoFB) and H9 human embryonic stem (hES) cells. These cell types are challenging to transfect and typically show low levels of homologous recombination. To determine the cell cycle phase that is optimal for HDR in neoFB cells, we used the same six chemical inhibitors to synchronize neoFB cells at G1, S and M phases of the cell cycle, followed by release prior to nucleofection with Cas9 RNP (***Figure 4A***). Cell cycle synchronization was confirmed by

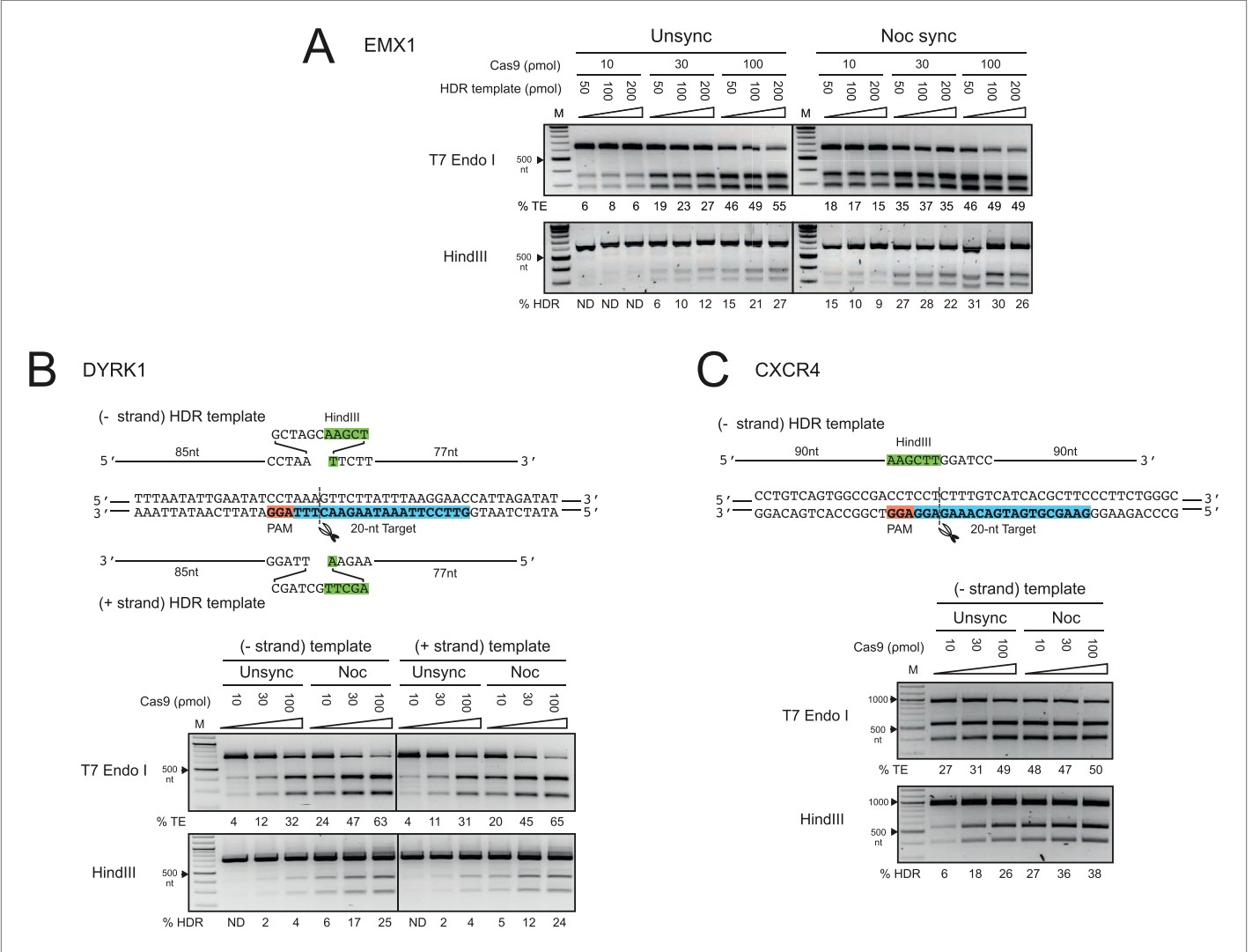

**Figure 2**. The enhancement of TE and HDR at the EMX1, DYRK1 and CXCR4 loci by nocodazole synchronization in HEK293T cells. (**A**) The effect of nocodazole on the TE and HDR frequencies at EMX1 locus. HEK293T cells were synchronized at M phase with 200 ng/ml of nocodazole for 17 hr before nucleofection. To determine the optimal dosage, three concentrations of Cas9 RNP were assayed in combination with three doses of HDR template (*Figure 1C*). The TE frequencies at 10 ρmol of Cas9 RNP in the unsynchronized cells were too low and therefore not determined (ND). (**B**) The effect of nocodazole on the TE and HDR frequencies at DYRK1 locus. The directionality of ssODNA HDR templates, either identical (+strand) or complementary (−strand) to the target sequence, was examined. The PAM is highlighted in red, the target sequence in blue and the integrated HindIII site in green. (**C**) The effect of nocodazole on the TE and HDR frequencies at the CXCR4 locus. The HDR template is ssODNA complementary (−strand) to the target sequence, and contains a HindIII restriction sequence flanked by 90 nt homology arms. Representative gels from two biological replicates are shown.

The following figure supplement is available for figure 2:

**Figure supplement 1**. On-target NHEJ and HDR and off-target cleavage analyses by deep sequencing.

FACS analysis, and cells in all conditions appeared morphologically normal. Aphidicolin-synchronized cells progressed normally through the cell cycle following release from cell cycle block (*Figure 1— figure supplement 1B*). In contrast to HEK293T cells, enhancement in TE and HDR frequencies was observed with aphidicolin and thymidine treatments, which synchronize the cells at S phase (*Figure 4A*). The TE frequencies were 17% and 13% with aphidicolin and thymidine respectively, as opposed to 5% in the unsynchronized condition. However, HDR frequency was very low across all conditions and was not detected in the unsynchronized cells. With aphidicolin synchronization, 0.6% and 1.3% HDR were detected at 30 and 100 ρmol Cas9 RNP respectively.

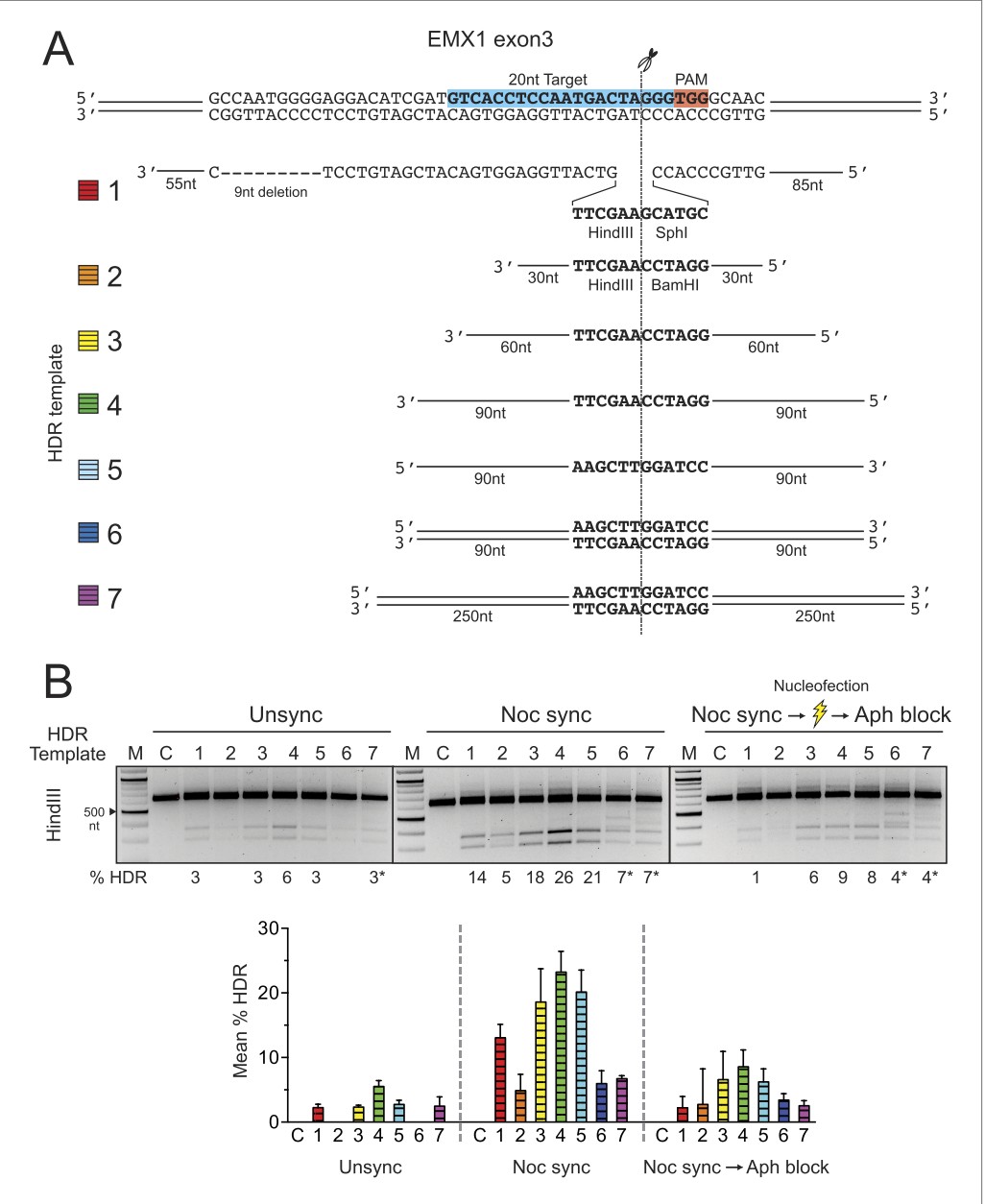

**Figure 3**. Systematic investigation of DNA templates for efficient HDR at the EMX1 locus in HEK293T cells.
(**A**) Segment of human EMX1 exon 3 shows the 20 nt target sequence (highlighted in blue), the TGG PAM region
(in red) and the Cas9 cleavage site at three bases upstream from PAM. Seven HDR templates (color coded) were tested
for HDR efficiency. Template 1 is as described in *Figure 1C*. Templates 2–7 contain HindIII and BamHI restriction sites
that are flanked symmetrically by various lengths of homology arms, ranging from 30 nt to 250 nt. Templates 2–5 are
ssODNA; templates 6–7 are PCR amplified double-stranded DNA (see 'Materials and methods'). (**B**) HDR efficiency
was tested under three cell conditions. In addition the unsynchronized and nocodazole synchronized conditions, the
cells were synchronized with nocodazole prior to nucleofection, and immediately post nucleofection, a single dose of
aphidicolin (2 μg/ml) was added to the growth media to prevent the transfected cells from proceeding into the S phase.
The purpose was to test whether blocking passage through S phase reduces HDR efficiency, since the HDR pathway is
thought to be most active during S phase. This one-time addition of aphidicolin is labeled as 'Aph block' in the third
panel, as opposed to the standard aphidicolin synchronization procedure used elsewhere in the manuscript. Thirty ρmol
of Cas9 RNP and 50 ρmol of HDR template were used in the nucleofection reaction; the control reaction (C) contained
no HDR template. The mean % HDR and standard deviation (error bar) was determined by HindIII digestion from
three experiments. Representative gels from PCR and HDR analyses are shown for each cell condition. Templates 6
and 7 produced unusual banding patterns, making quantitation of DNA bands less accurate (labeled by asterisk).

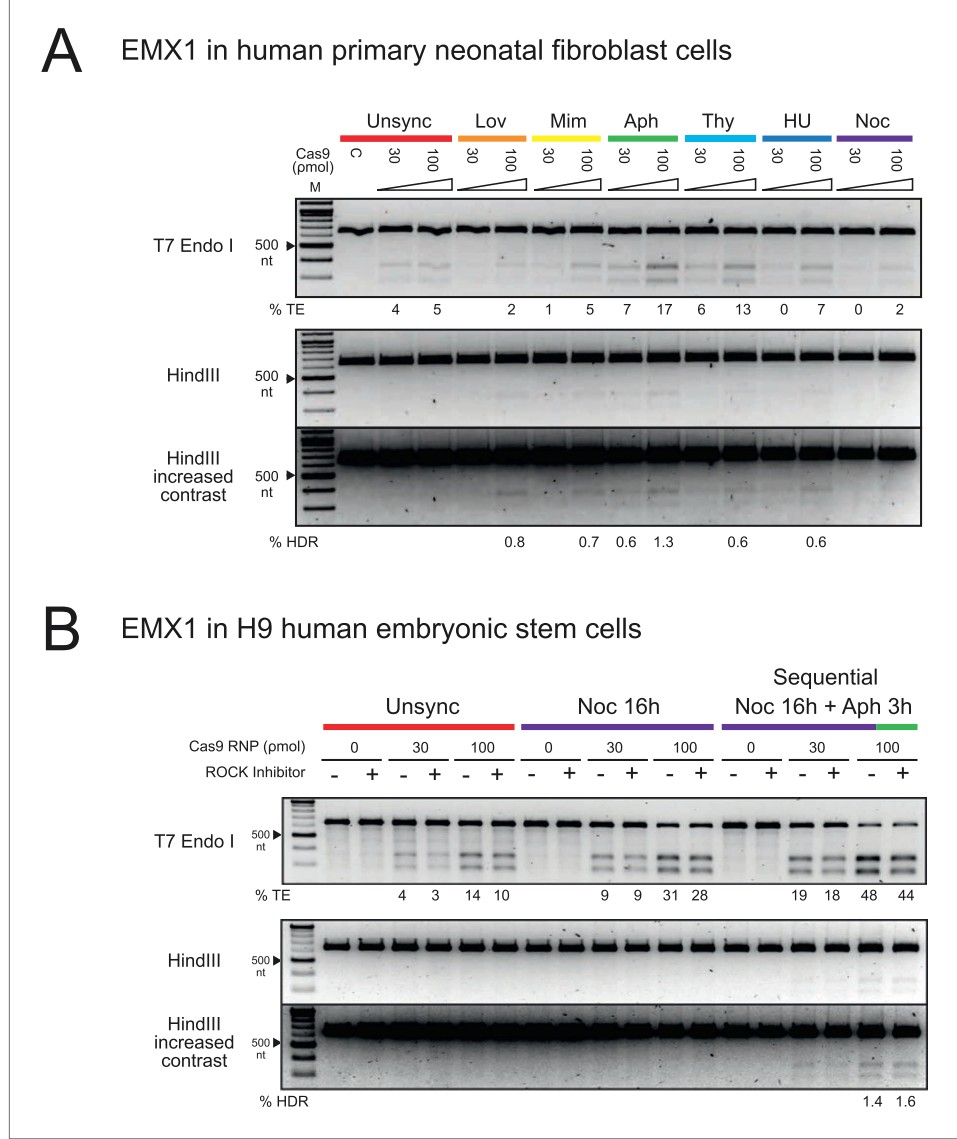

**Figure 4**. The enhancement of TE and HDR frequencies at the EMX1 locus by cell cycle synchronization in human primary neonatal fibroblast and embryonic stem cells. (**A**) Screening of cell cycle inhibitors for enhancement of TE and HDR frequencies in human primary neonatal fibroblast cells. For each inhibitor condition (color coded), two doses of Cas9 RNP, 30 and 100 ρmol, were transfected with 100 ρmol of HDR DNA template 4 from *Figure 3A*. A control reaction (labeled as C) contained 100 ρmol of Cas9 but no sgRNA. The % TE and % HDR were analyzed similarly as with HEK293T cells. (**B**) Three conditions were tested using hES cells: unsynchronized, nocodazole synchronized and nocodazole-aphidicolin sequential synchronized. The cells were treated with nocodazole for 16 hr, washed to remove the drug and then treated with aphidicolin for 3 hr before nucleofection. 30 or 100 ρmol of Cas9 RNP was co-transfected with 100 ρmol of HDR template 4 from *Figure 3A*, and cultured at high density in 96-well plates in the presence or absence of ROCK apoptosis inhibitor (10 μM). For both experiments, representative gels from two biological replicates are shown. The contrast of the gel images was increased to show that no HDR was detected in other conditions.

We then tested hES cells in a similar experimental setup. Previous reports have shown ~20% NHEJ with transfection of Cas9 RNPs but didn't analyze HDR rates (*Kim et al., 2014*). Also, HDR frequencies are low with nucleic acid-based delivery of Cas9 (*Hsu et al., 2013*). Screening of chemical inhibitors showed that only nocodazole synchronization enhanced TE frequencies in hES cells to 9% at 30 ρmol and 28–31% at 100 ρmol Cas9 RNP; however, we did not detect HDR with the HDR template (−) sense strand (*Figure 4B*). In light of these observations, we modified a protocol from

Pauklin et al. (*Pauklin and Vallier, 2013*), in which the cells were treated with nocodazole for 16 hr, washed to remove the drug, and then treated with aphidicolin for 3 hr before nucleofection. Using this approach, we detected ~2% of HDR at 100 pmol Cas9 RNP (*Figure 4B*). Cell synchronization was confirmed by FACS analysis. 3 days after release from cell cycle synchronization the hES cell cycle behavior was indistinguishable from unsynchronized control cultures, with no apparent changes in colony morphology (*Figure 1—figure supplement 1C*); all colonies expressed high levels of alkaline phosphatase, a marker for pluripotency (*Figure 1—figure supplement 1D*). ROCK apoptosis inhibitor (10 μM) was required for survival of synchronized H9 ESCs after release when cultured at low density but not when cultured at high density. These results suggest that different cell types will have distinct requirements for synchronization to enhance Cas9 RNP-induced DNA repair. Also, it may be possible to find conditions that will enable even higher levels of HDR in neoFB and hES cells using Cas9 RNP delivery.

## Discussion

Here we report a simple and robust system to enhance genome engineering by HDR in human cells using cell cycle synchronization and timed delivery of Cas9 ribonucleoprotein complexes. Advantages of this approach include no detectable off-target editing, timed introduction of pre-assembled editing complexes into cells and simultaneous transfection of multiple Cas9 RNPs and donor DNAs. In addition, Cas9 RNP-mediated editing begins within 4 hr of delivery and is largely completed within 24 hr due to RNP degradation (*Kim et al., 2014*). Furthermore, higher cell viability has been observed following RNP transfection compared with DNA transfection (*Kim et al., 2014*; *Zuris et al., 2014*). These features enable robust levels of on-target editing while reducing off-target effects.

Using this system, we have maximized the efficiency of HDR such that ~33% of detected DSB repair events occur with homologous recombination of donor DNA. We chose the EMX1 target sequence to compare with published results using nucleic acid delivery, for which 10% HDR efficiency was reported in HEK293T cells (*Ran et al., 2013*). Our results are also significantly higher than reported rates of HDR in synchronized HCT116 cells using Transcription Activator-Like Effector Nucleases (TALENs) and higher than typically observed using nucleic acid-based delivery of Cas9 (*Rivera-Torres et al., 2014*). (*Hsu et al., 2013*). Further increase of HDR efficiency beyond 33% will likely require manipulation of the proteins involved in the HDR or NHEJ pathways (*Humbert et al., 2012*).

It is surprising that nocodazole treatment leads to higher HDR efficiency at reduced dosage of Cas9 RNPs. Nocodazole blocks cells at M phase when the DNA is fully replicated and the nuclear membrane is broken down. One explanation may be that delivery of Cas9 RNPs into a nocodazole-synchronized cell effectively targets two cells because they divide upon release. Another possibility is that once the nuclear envelope is broken down, Cas9 RNPs can gain easy access to the DNA. The resulting high HDR frequencies without off-target editing provide an important advance for generating scar-less genetic modifications, including epitope-tagged alleles, reporter genes, precise insertions and deletions and point mutations. Together these results expand the utility of CRISPR/Cas9-mediated genome engineering in human cells and provide a foundation for further advances using Cas9 RNP delivery methods.

## Materials and methods

### Cell lines and cell culture

DMEM media, non-essential amino acid, penicillin-streptomycin, E8 media, DPBS and 0.05% trypsin were purchased from Life Technologies, Carlsbad, CA. HEK293T cells and human neonatal dermal fibroblasts (catalog #2310: ScienCell, Carlsbad, CA) were maintained in DMEM media supplemented with 10% fetal bovine serum, non-essential amino acid and penicillin-streptomycin. H9 human embryonic stem cells were maintained on Matrigel (Corning, Tewksbury, MA) in E8 media plus supplement (Life Technologies, Carlsbad, CA).

### Cell cycle synchronization

Aphidicolin, hydroxyurea, lovastatin, mimosine, nocodazole and thymidine were purchased from Sigma–Aldrich, St. Louis, MO. The synchronization protocols were modified from the following references (*Adams and Lindsay, 1967*; *Harper, 2007*; *Jackman and O'Connor, 2001*; *Pauklin and Vallier, 2013*). It is important to ensure cells are maintained at <70% confluency. HEK293T cells were seeded at low density, $3 \times 10^6$ cell density in a 10-cm culture dish and human primary neonatal fibroblasts seeded

at $1.2 \times 10^6$ in a 15-cm dish 17 hr before transfection. Aphidicolin and thymidine require two sequential treatments to enrich cells arrested at the entry of S phase (*Jackman and O'Connor, 2001*). Cells were treated with aphidicolin (2 µg/ml) or thymidine (5 mM) for 17 hr, washed with media to remove the drugs, grown for 8 hr, and treated with a second dose of drugs for 17 hr. In the experiment in *Figure 3B*, third panel, a single dose of aphidicolin (2 µg/ml) was added to the nocodazole synchronized cells immediately after nucleofection. Hydroxyurea (2 mM), lovastatin (40 µM), mimosine (200 µM) and nocodazole (200 ng/ml) require only one treatment for 17 hr. Two synchronization conditions were tested in the human ES cell experiment as shown in *Figure 4B*. Human ES cells were cultured in 6 well dishes, split 1:10 3 days before adding nocodozole. The first condition was a simple nocodazole treatment for 16 hr. The second condition was modified from *Pauklin and Vallier, 2013*. The cells were treated with nocodazole for 16 hr, washed to remove the drug, and then treated with aphidicolin for 3 hr before nucleofection. We shortened the duration of aphidicolin treatment, because we noticed a substantial drop in cell viability at 10 hr. After transfection cells were either seeded at high density in a 96 well plate for analysis of editing or low density, 6-well plate, for imaging and long term growth.

### Cell cycle analysis

The cell cycle analysis was performed using BD Biosciences (San Jose, CA) BrdU-FITC FACS kit, to determine the percent of cells in each phase of the cell cycle. HEK293T and H9 ES cells were incubated with BrdU for 45 min while Fibroblasts were incubated with BrdU for 2 hr. To determine the percent of cells in G2/M, DNA was stained with 7-AAD (7-aminoactinomycin D) and analyzed on a BD Fortessa Flow Cytometer.

### Alkaline phosphatase staining

Followed Millipore (Billerica, MA) Alkaline Phosphatase detection kit protocol, Cat. No. SCR004.

### Expression and purification of Cas9

The recombinant *S. pyogenes* Cas9 used in this study carries at C-terminus an HA tag and two nuclear localization signal peptides which facilitates transport across nuclear membrane. The protein was expressed with a N-terminal hexahistidine tag and maltose binding protein in *E. coli* Rosetta 2 cells (EMD Millipore, Billerica, MA) from plasmid pMJ915. The His tag and maltose binding protein were cleaved by TEV protease, and Cas9 was purified by the protocols described in *Jinek et al., 2012*. Cas9 was stored in 20 mM 2-[4-(2-hydroxyethyl)piperazin-1-yl]ethanesulfonic acid (HEPES) at pH 7.5, 150 mM KCl, 10% glycerol, 1 mM tris(2-chloroethyl) phosphate (TCEP) at −80°C.

### In vitro T7 transcription of sgRNA

The DNA template encoding for a T7 promoter, a 20 nt target sequence and an optimized sgRNA scaffold (*Chen et al., 2013*) was assembled from synthetic oligonucleotides (Integrated DNA technologies, San Diego, CA) by overlapping PCR. Briefly, for the EMX1 sgRNA template, the PCR reaction contains 20 nM premix of BS16 (5′- TAA TAC GAC TCA CTA TAG GTC ACC TCC AAT GAC TAG GGG TTT AAG AGC TAT GCT GGA AAC AGC ATA GCA AGT TTA AAT AAG G -3′) and BS6 (5′- AAA AAA AGC ACC GAC TCG GTG CCA CTT TTT CAA GTT GAT AAC GGA CTA GCC TTA TTT AAA CTT GCT ATG CTG TTT CCA GC -3′), 1 µM premix of T25 (5′- TAA TAC GAC TCA CTA TAG -3′) and BS7 (5′- AAA AAA AGC ACC GAC TCG GTG C -3′), 200 µM dNTP and Phusion Polymerase (NEB, Ipswich, MA) according to manufacturer's protocol. The thermocycler setting consisted of 30 cycles of 95°C for 10 s, 57°C for 10 s and 72°C for 10 s. The PCR product was extracted once with phenol:chloroform:isoamylalcohol and then once with chloroform, before isopropanol precipitation overnight at −20°C. The DNA pellet was washed three times with 70% ethanol, dried by vacuum and dissolved in DEPC-treated water. The DYRK1 sgRNA template was assembled from T25, BS6, BS7 and BS14 (5′- TAA TAC GAC TCA CTA TAG GTT CCT TAA ATA AGA ACT TTG TTT AAG AGC TAT GCT GGA AAC AGC ATA GCA AGT TTA AAT AAG G -3′). The CXCR4 sgRNA template was assembled from T25, SLKS3 (5′- TAA TAC GAC TCA CTA TAG GAA GCG TGA TGA CAA AGA GGG TTT TAG AGC TAT GCT GGA AAC AGC ATA GCA AGT TAA AAT AAG G -3′), SLKS1 (5′- GCA CCG ACT CGG TGC CAC TTT TTC AAG TTG ATA ACG GAC TAG CCT TAT TTT AAC TTG CTA TGC TGT TTC AGC -3′) and SLKS2 (5′- GCA CCG ACT CGG TGC CAC TTT TTC AAG -3′).

An 100-µl T7 in vitro transcription reaction consisted of 30 mM Tris–HCl (pH 8), 20 mM MgCl$_2$, 0.01% Triton X-100, 2 mM spermidine, 10 mM fresh dithiothreitol, 5 mM of each ribonucleotide triphosphate, 100 µg/ml T7 Pol and 1 µM DNA template. The reaction was incubated at 37°C for 4 hr,

and 5 units of RNase-free DNaseI (Promega, Madison, WI) was added to digest the DNA template 37°C for 1 hr. The reaction was quenched with 2xSTOP solution (95% deionized formamide, 0.05% bromophenol blue and 20 mM EDTA) at 60°C for 5 min. The RNA was purified by electrophoresis in 10% polyacrylamide gel containing 6 M urea. The RNA band was excised from the gel, grinded up in a 15-ml tube, and eluted with 5 vol of 300 mM sodium acetate (pH 5) overnight at 4°C. One equivalent of isopropanol was added to precipitate the RNA at −20°C. The RNA pellet was collected by centrifugation, washed three times with 70% ethanol, and dried by vacuum. To refold the sgRNA, the RNA pellet was first dissolved in 20 mM HEPES (pH 7.5), 150 mM KCl, 10% glycerol and 1 mM TCEP. The sgRNA was heated to 70°C for 5 min and cooled to room temperature. $MgCl_2$ was added to a final concentration of 1 mM. The sgRNA was again heated to 50°C for 5 min, cooled to room temperature and kept on ice. The sgRNA concentration was determined by $OD_{260nm}$ using Nanodrop and adjusted to 100 µM using 20 mM HEPES (pH 7.5), 150 mM KCl, 10% glycerol, 1 mM TCEP and 1 mM $MgCl_2$. The sgRNA was store at −80°C.

## PCR assembly of HDR template 6 and 7

Double-stranded HDR template 6 and 7 were prepared by PCR amplification. Template 6 was PCR amplified from single-stranded template 5 (5′- TGG CCA GGG AGT GGC CAG AGT CCA GCT TGG GCC CAC GCA GGG GCC TGG CCA GCA GCA AGC AGC ACT CTG CCC TCG TGG GTT TGT GGT TGC GGA TCC AAG CTT TTG GAG GTG ACA TCG ATG TCC TCC CCA TTG GCC TGC TTC GTG GCA ATG CGC CAC CGG TTG ATG TGA TGG GAG CCC TTC TTC TTC TGC TCG -3′) using primer set (forward 5′- CGA GCA GAA GAA GAA GGG CTC CCA TC -3′ and reverse 5′- TGG CCA GGG AGT GGC CAG AGT CC -3′). The PCR reaction was performed using Phusion Polymerase according to manufacturer's protocol (NEB, Ipswich, MA). The thermocycler setting consisted of 30 cycles of 95°C for 20 s, 67°C for 10 s and 72°C for 20 s. The PCR product was extracted once with phenol:chloroform:isoamylalcohol and then once with chloroform, before isopropanol precipitation overnight at −20°C. The DNA pellet was washed three times with 70% ethanol, dried by vacuum and dissolved in water. The concentration was determined by Nanodrop (Thermo Fisher Scientific, Waltham, MA).

Template 7 was assembled from two fragments (A and B) by overlapping PCR. Fragment A was PCR amplified from HEK293T genomic DNA using the primer set (forward 5′- GCT CAG CCT GAG TGT TGA GGC CCC AGT GGC TGC TCT GG -3′ and reverse 5′- GTG GTT GCG GAT CCA AGC TTT TGG AGG TGA CAT CGA TGT CCT CCC CAT GGG C -3′). Fragment B was amplified using the primer set (forward 5′- CAC CTC CAA AAG CTT GGA TCC GCA ACC ACA AAC CCA CGA GGG CAG AGT GCT GCT TGC -3′ and reverse 5′- TGC GGT GGC GGG CGG GCC CGC CCA GGC AGG CAG GC -3′). Both reaction were performed using Kapa Hot start high-fidelity polymerase (Kapa Biosystems, Wilmington, MA) in high GC buffer according to the manufacturer's protocol. The thermocycler setting consisted of one cycle of 95°C for 5 min, 30 cycles of 98°C for 20 s, 67°C for 10 s and 72°C for 20 s, and one cycle of 72°C for 1 min.

## Cas9 RNP assembly and nucleofection

Cas9 RNP was prepared immediately before experiment by incubating with sgRNA at 1:1.2 molar ratio in 20 mM HEPES (pH 7.5), 150 mM KCl, 1 mM $MgCl_2$, 10% glycerol and 1 mM TCEP at 37°C for 10 min. HDR template was then added to the RNP mixture. Cells were dissociated by 0.05% trypsin, spun down by centrifugation at 400×g for 3 min, and washed once with DPBS. Nucleofection of HEK293T cells was performed using Lonza (Allendale, NJ) SF cell- kits and program CM130 in an Amaxa 96-well Shuttle system. The human neoFB were transfected with Lonza P2 kit and program CA137. The hES cells were transfected with P3 primary cell kit and program CB150. Each nucleofection reaction consisted of approximately $2 \times 10^5$ cells in 20 µl of nucleofection reagent and mixed with 10 µl of RNP:DNA. After electroporation, 100 µl of growth media was added to the well to transfer the cells to tissue culture plates. The cells were incubated at 37°C for 24 hr, the media was removed by aspiration, and 100 µl of Quick Extraction solution (Epicentre, Madison, WI) was added to lyse the cells and extract the genomic DNA. The cell lysate was incubated at 65°C for 20 min and then 95°C for 20 min, and stored at −20°C. The concentration of genomic DNA was determined by NanoDrop (Thermo Fisher Scientific, Waltham, MA).

## PCR amplification of target region

A 640 nt region of EMX1 and DYRK1 loci, containing the target site, were PCR amplified using the following primer sets. For EMX1: forward 5′- GCC ATC CCC TTC TGT GAA TGT TAG AC -3′ and

5'- GGA GAT TGG AGA CAC GGA GAG CAG -3'. For DYRK1: forward 5'- GAG GAG CTG GTC TGT TGG AGA AGT C -3' and reverse 5'- CCC AAT CCA TAA TCC CAC GTT GCA TG -3'. A 903 nt region of CXCR4 locus was amplified using primer set: 5'- AGA GGA GTT AGC CAA GAT GTG ACT TTG AAA CC -3' and 5'- GGA CAG GAT GAC AAT ACC AGG CAG GAT AAG GCC -3'. These primers were designed to avoid amplifying the HDR templates by annealing outside of the homology arms. The PCR reaction was performed using 200 ng of genomic DNA and Kapa Hot start high-fidelity polymerase (Kapa Biosystems, Wilmington, MA) in high GC buffer according to the manufacturer's protocol. The thermocycler setting consisted of one cycle of 95°C for 5 min, 30 cycles of 98°C for 20 s, 62°C for 15 s and 72°C for 30 s, and one cycle of 72°C for 1 min. The PCR products were analyzed on 2% agarose gel containing SYBR Safe (Life Technologies, Carlsbad, CA). The concentration of PCR DNA was quantitated based on the band intensity relative to a DNA standard using the software Image Lab (Bio-Rad, Hercules, CA). About 200 ng of PCR DNA was used for T7 endonuclease I and HindIII analyses.

### Analysis of %TE by T7 endonuclease I assay

The percentage of Cas9 induced TE was determined by T7 endonuclease I assay. T7 endonuclease I recognizes and cleaves mismatched heteroduplex DNA which arises from hybridization of wild-type and mutant DNA strands. The hybridization reaction contained 200 ng of PCR DNA in KAPA high GC buffer and 50 mM KCl, and was performed on a thermocycler with the following setting: 95°C, 10 min, 95–85°C at −2°C/sec, 85°C for 1 min, 85–75°C at −2°C/sec, 75°C for 1 min, 75–65°C at −2°C/sec, 65°C for 1 min, 65–55°C at −2°C/sec, 55°C for 1 min, 55–45°C at −2°C/sec, 45°C for 1 min, 45–35°C at −2°C/sec, 35°C for 1 min, 35–25°C at −2°C/sec, 25°C for 1 min, and hold at 4°C. Buffer 2 and 5 units of T7 endonuclease I (NEB, Ipswich, MA) were added to digest the re-annealed DNA. After 1 hr of incubation at 37°C, the reaction was quenched with one volume of gel loading dye (50 mM Tris pH 8.5, 50 mM EDTA, 1% SDS, 50% glycerol and 0.01% bromophenol blue) at 70°C for 10 min. The product was resolved on 2% agarose gel containing SYBR gold (Life technologies, Carlsbad, CA). The DNA band intensity was quantitated using Image Lab. The TE frequency was measured using a T7 endonuclease I assay and calculated using the following equation $(1 - (1 - (b + c / a + b + c))^{1/2}) \times 100$, where 'a' is the band intensity of DNA substrate and 'b' and 'c' are the cleavage products (*Ran et al., 2013*). Using this formula is necessary, because upon re-annealing, one duplex of mutant DNA can produce two duplexes of mutant:wild-type hybrid, doubling the actual TE frequency.

### Analysis of HDR by HindIII restriction digestion

HindIII directly cleaves PCR DNA containing the newly integrated HindIII restriction sequence as the result of successful HDR. The reaction consisted of 200 ng of PCR DNA and 10 units of HindIII High Fidelity in CutSmart Buffer (NEB, Ipswich, MA). After 2 hr of incubation at 37°C, the reaction was quenched with one volume of gel loading dye at 70°C for 10 min. The product was resolved on 2% agarose gel containing SYBR gold (Life technologies, Carlsbad, CA). The band intensity was quantitated using Image Lab. The percentage of HDR was calculated using the following equation $(b + c / a + b + c) \times 100$, where 'a' is the band intensity of DNA substrate and 'b' and 'c' are the cleavage products.

### Deep sequencing analysis of on-target and off-target sites

The genomic region flanking the CRISPR target site for each gene was amplified by 2-step PCR method using primers listed in *Supplementary file 1*. First, the genomic DNA from the edited and control samples was isolated and PCR amplified 15 cycles using Kapa Hot start high-fidelity polymerase (Kapa Biosystems, Wilmington, MA) according to the manufacturer's protocol. The resulting amplicons were purified by AMPure beads to remove primers and subjected to five cycles of PCR to attach Illumina P5 adapters as well as unique sample-specific barcodes followed by bead purification. Berkeley Sequencing facility performed the AMPure bead cleanup. Barcoded and purified DNA samples were quantified by Qubit 2.0 Fluorometer (Life Technologies, Carlsbad, CA), size analyzed by BioAnalyzer, quantified by qPCR and pooled in an equimolar ratio. Sequencing libraries were sequenced with the Illumina MiSeq Personal Sequencer (Life Technologies, Carlsbad, CA).

Amplicon sequencing data were analyzed as described below. The 300-bp paired end MiSeq raw reads were de-multiplexed using Illumina MiSeq Reporter software. This generated sample specific paired end raw read files (R1 and R2 fastq files). Adapter and windowed adaptive quality trimming was performed on the raw reads (using Trim Galore). Reads containing bases with a PHRED quality score of less than 30 were removed. R1 and reverse complemented R2 reads were then merged into sample

specific fasta file. Smith Waterman alignments (EMBOSS Water) were performed for each sample reads against the corresponding 53 nucleotide reference locus. These 53 nt for each locus included the 23 nt target sequence with 15 nt flanking sequences. Alignments were filtered to assess presence of indels and homologous recombination. Reads were considered to have indels if their alignments were at least 53 nt long and had any gaps. Reads were considered non-indels if their alignments were at least 53 nucleotides long without any gaps. TE frequency was calculated as 100 x #indel reads/ (#indel reads + #non-indel reads). Reads were considered to have homologous recombination if alignments were at least 53 nucleotides long and had a AAGCTTGCTAGC insertion for the EMX1 loci (both on and off target) and a GCTAGCAAGCTT insertion for the DYRK1 loci (both on and off target). HDR frequency was calculated as either 100 x #HDR reads/(#indel reads + #non-indel reads). Deep sequencing data is available at the NCBI Sequence Read Archive (SRA, BioProject: 269153).

## Additional information

### Funding

| Funder | Grant reference number | Author |
|---|---|---|
| Damon Runyon Cancer Research Foundation (Damon Runyon) | DRG2176-13 | Steven Lin |
| Roche | Postdoctoral Fellowship, RPF311 | Brett T Staahl |

The funders had no role in study design, data collection and interpretation, or the decision to submit the work for publication.

### Author contributions

SL, BTS, Conception and design, Acquisition of data, Analysis and interpretation of data, Drafting or revising the article; RKA, Analysis and interpretation of data; JAD, Conception and design, Analysis and interpretation of data, Drafting or revising the article

## Additional files

### Supplementary file

• Supplementary file 1. PCR primers for target loci amplification and indexing for deep sequencing of on-target and off-target sites. Also included is deep sequencing data and analysis.

### Major dataset

The following dataset was generated:

| Author(s) | Year | Dataset title | Dataset ID and/or URL | Database, license, and accessibility information |
|---|---|---|---|---|
| Lin S, Staahl B, Alla RK, Doudna JA | 2014 | SRA BioProject: Accession number SRX806264 | | Publicly available at NCBI Sequence Read Archive. |

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
