## [Decision Letter]

Thank you for sending your work entitled “Enhanced homology-directed human genome
engineering by controlled timing of CRISPR/Cas9 delivery” for consideration at
*eLife*. Your article has been favorably evaluated by Detlef Weigel
(Senior editor) and 2 reviewers, whose detailed comments are attached at the bottom of
this email. All three discussed their comments before we reached this decision. The
following comments will help you prepare a revised submission.

While an effect of the cell cycle on gene targeting has been shown before, this is the
first such demonstration for the CRISPR/Cas9 system, and the overall agreement was that
it is an important extension of your previous study that will be of great utility to the
community at large. Before we can accept the paper for publication, we ask that you
specifically address the following three major concerns:

1) Characterize in more detail the effects of the cell cycle inhibitors on HEK293T and
H9 ES cells, and of nucleofection versus DNA transfection.

2) Preferably add information on other cell lines (or clearly state caveat that
different optimization might be needed for each cell line, including nucleofection
efficiencies).

3) Demonstrate the superiority of the method over the conventional approach by targeting
additional sites in the genome.

Reviewer #1:

The CRISPR/Cas9 system is by now a widely used system for site-directed genome editing.
Upon site directed cleavage by an RNA-guided CRISPR/Cas9 protein a double strand break
(DSB) is introduced. The DSB can be sealed either by error-prone non-homologous end
joining (NHEJ) or homology-directed repair (HDR). Typically NHEJ is strongly favored in
cells and thus integration of desired DNA into a given genome location by HDR is
difficult to achieve. Accordingly, people have tried in the past to identify
experimental parameters that can be modified to increase HDR efficiency.

The article under review is very well written and the figures are well designed and
clear. However, the Results section of this article contains many numbers. Maybe the
authors can identify sections where they can make general statements rather than only
listing all data points. The text in its current form is hard to read due to the
excessive listing of data.

The manuscript under review focused on optimization of the experimental parameters of
CRISPR/Cas9-mediated genome editing in human cells to achieve higher HDR frequencies.
The authors take advantage of the fact that the ratio of HDR versus NHEJ differs in
different cell cycles. They use a set of chemicals that induce arrest within defined
cell cycles. As mentioned by the authors, this idea has been used before in combination
with TAL-effector based nucleases (TALENs) (Rivera-Torres , 2014, PLoS ONE 9: e96483).
However, it seems that TALENs work is less efficient in combination with cell-cycle
arresting chemicals as CRISPR/Cas9 nucleases. The judgment of the authors seems correct.
Yet, the fact remains that the conceptual idea is not novel but has been published
before.

In addition to the use of cell-cycle arresting chemicals, the authors tested other
experimental parameters (e.g. concentration of CRISPR/Cas9, guide RNA and length of the
HDR repair template) to improve HDR efficiency. In their studies, they identify various
parameters like the CRISPR/Cas9 concentration, which significantly improve HDR
efficiency. They also provide data indicating that experimental conditions that improve
HDR do not cause increased off-target activity. I believe that the identified
experimental conditions will be of use for the huge community of scientists that want to
use the CRISPR/Cas9 system for HDR in human cells. Yet, while the improvements are
typically significant they remain in most cases marginal. Furthermore, the authors did
not test if the used chemicals will have non-desirable side-effects (e.g. high overall
mutation rate, increased cell-death frequencies, etc.). I assume the cellular
consequences that are linked to the use of these chemicals are in many cases known and
thus should be tested experimentally. It should be straightforward to test at least the
effect on viability of treated cells. I am convinced that these studies would increase
the value of the studies.

I believe the article will be of use to a broad community. Yet, the article provides no
fundamental or novel mechanistical insights, and even the central idea of using
cell-cycle arresting chemicals is not conceptually novel. In sum, a useful article for
applied biotech but no news for fundamental biology.

Reviewer #2:

The CRISPR/Cas9 system is rapidly revolutionizing the field of genome engineering,
allowing researchers to manipulate both coding and non-coding genomic sequences at will
in a constantly growing number of biological systems. This system creates double-strand
breaks (DSBs) at target loci, which can be repaired through one of two cellular
mechanisms: non-homologous end joining (NHEJ) or homology-directed repair (HDR). The
ability of a cell to repair a DSB generated by Cas9 through HDR-mediated incorporation
of exogenous DNA templates has recently been exploited to engineer several modifications
to endogenous loci, such as novel knock-in alleles, point mutations and fluorescent
tags, among others. However, the frequency of NHEJ is usually higher than HDR due to the
fact that NHEJ does not require any homologous or exogenous DNA molecules to repair the
DSB. Therefore, developing experimental methods that increase the frequency of HDR is
important in order for this technology to fulfill its full potential in various
laboratory studies as well as in clinical applications.

Lin et al. have addressed whether cell cycle synchronization might affect the relative
use of these two repair pathways in an effort to define conditions that lead to more
efficient HDR. The authors tested whether reversible treatment of cells with drugs
reported to arrest cells in the S and late G2 cell cycle phases could increase the rate
of HDR when combined with timed delivery of Cas9-sgRNA ribonucleoprotein complexes
(RNPs) and various exogenous DNA templates. Using six pharmacological agents and further
narrowing the list down to two (nocodazole, which is reported to block cells at late
G2/M phase, and aphidicolin, which blocks cells at S phase), the authors convincingly
demonstrate that these treatments, coupled with timed delivery of Cas9 RNPs and
exogenous DNA templates, significantly increased the rate of HDR across two loci in two
different cell types. Importantly, they also convincingly demonstrate that off-target
editing is negligible using this approach.

This is a significant extension of this groups previous efforts aimed at establishing
the CRISPR/Cas9 genome editing system in mammalian cells for inducing both NHEJ and HDR
at specific genomic loci (13).
The ability to increase the rate of HDR through cell synchronization coupled with timed
delivery of Cas9 RNPs will undoubtedly have a significant impact in the field of genome
editing, particularly for applications aimed at engineering specific mutations of
interest into a variety of cell types, such as human ES and iPS cells.

Major comments:

1) The key observation here is that cells treated and released from different chemical
inhibitors of cell cycle progression undergo increased CRISPR/Cas9-mediated HDR compared
to untreated cells. While this is clearly shown, the cell cycle effects that are
associated with this treatment are not well characterized. Although the drugs employed
are commonly used in the field, it is important to characterize them in the particular
cell lines studied. The authors show cell cycle analysis in of HEK293T cells in Figure 1—figure supplement 1. However, for
both nocodazole and aphidicolin treatment, the data appear to show significant 2n as
well as 4n peaks. Thus, it is unclear what cell cycle phase might be associated with the
increased HDR observed following release from these treatments. Regarding the
experiments with H9 human ES cells, in which a combination of nocodazole and aphidicolin
was used, there is no cell cycle analysis shown at all. It will be important to address
both of these issues prior to publication.

2) With the exception of the experiments presented in Figure 2 using the H9 cells, most of the experiments were carried out with
HEK293T cells, which are readily transfectable with nucleofection methods. Importantly,
the rate of HDR reported was significantly lower in drug-treated H9 cells compared to
drug-treated HEK293T cells. Moreover, induction of HDR in ES cells required a
modification of the protocol to incorporate a 16-hour pulse of nocodazole followed by a
3-hour pulse of aphidicolin before Cas9 RNP nucleofection. One wonders how generalizable
these methods will be to other cell types. Therefore, the manuscript would be
strengthened with the addition of analysis of a panel of cell lines.

3) The authors should establish the baseline nucleofection efficiencies for the
different cell lines tested. This will help clarify whether nucleofection efficiency
many be a contributing factor in the difference seen between HEK293T cells and H9
cells.

4) It is unclear whether the other cell cycle inhibitors besides nocodazole shown in
Figure 1—figure supplement 1 were
tested in ES cells. Minimally, this point should be clarified. If they were not tested,
is there a reason why not?

5) Given the interest in targeting efficiencies as a function of target loci, it would
be useful to extend this study to more than the two loci tested here.

6) In Figure 3, the authors show that adding
aphidicolin following release from a nocodazole block reduced HDR efficiency in HEK293T
cells, suggesting that S-phase entry may be required for efficient HRD-mediated repair.
They should show that this combined treatment actually did block S-phase entry in these
experiments, especially given the odd cell cycle profiles shown in Figure 1—figure supplement 1. Also, how does this
conclusion jibe with the increased efficiency of HDR in ES cells treated with this same
combination when compared with nocodazole alone?

7) The authors argue in the Discussion that their approach of nucleofection of Cas9 RNPs
leads to higher cell viability than DNA transfection-based methods. However, no data is
shown to support this claim.

---

## [Author Response]

Reviewer #1:

*The CRISPR/Cas9 system is by now a widely used system for site-directed genome
editing. Upon site directed cleavage by an RNA-guided CRISPR/Cas9 protein a double
strand break (DSB) is introduced. The DSB can be sealed either by error-prone
non-homologous end joining (NHEJ) or homology-directed repair (HDR). Typically NHEJ
is strongly favored in cells and thus integration of desired DNA into a given genome
location by HDR is difficult to achieve. Accordingly, people have tried in the past
to identify experimental parameters that can be modified to increase HDR
efficiency*.

*The article under review is very well written and the figures are well designed
and clear. However, the Results section of this article contains many numbers. Maybe
the authors can identify sections where they can make general statements rather than
only listing all data points. The text in its current form is hard to read due to the
excessive listing of data*.

We agree with the reviewer that excessive listing of numerical data makes reading
difficult, and have now clarified the manuscript by removing several data listings and
keeping only the numbers that are essential to the discussion.

*The manuscript under review focused on optimization of the experimental
parameters of CRISPR/Cas9-mediated genome editing in human cells to achieve higher
HDR frequencies. The authors take advantage of the fact that the ratio of HDR versus
NHEJ differs in different cell cycles. They use a set of chemicals that induce arrest
within defined cell cycles. As mentioned by the authors, this idea has been used
before in combination with TAL-effector based nucleases (TALENs) (Rivera-Torres,
2014, PLoS ONE 9: e96483). However, it seems that TALENs work is less efficient in
combination with cell-cycle arresting chemicals as CRISPR/Cas9 nucleases. The
judgment of the authors seems correct. Yet, the fact remains that the conceptual idea
is not novel but has been published before*.

*In addition to the use of cell-cycle arresting chemicals, the authors tested
other experimental parameters (e.g. concentration of CRISPR/Cas9, guide RNA and
length of the HDR repair template) to improve HDR efficiency. In their studies, they
identify various parameters like the CRISPR/Cas9 concentration, which significantly
improve HDR efficiency. They also provide data indicating that experimental
conditions that improve HDR do not cause increased off-target activity. I believe
that the identified experimental conditions will be of use for the huge community of
scientists that want to use the CRISPR/Cas9 system for HDR in human cells. Yet, while
the improvements are typically significant they remain in most cases marginal.
Furthermore, the authors did not test if the used chemicals will have non-desirable
side-effects (e.g. high overall mutation rate, increased cell-death frequencies,
etc.). I assume the cellular consequences that are linked to the use of these
chemicals are in many cases known and thus should be tested experimentally. It should
be straightforward to test at least the effect on viability of treated cells. I am
convinced that these studies would increase the value of the studies*.

We thank the reviewer for offering critical evaluation of our work and we appreciate the
potential for side effects when using cell cycle inhibitors. First, we would like to
emphasize that the enhancement in editing efficiency is not marginal. We chose to
present the direct measure of %HDR, which is capped at 100%, instead of converting the
percent readouts into percent increase. Although in some cases the enhancement may
appear modest, these changes in %HDR could make a significant difference when deciding
to proceed with single cell isolation to obtain homozygous clones.

Second, we now include cell cycle analysis of the HEK293T cells, fibroblasts and hES
cells following release from cell cycle inhibitors in Figure 1—figure supplement 1. In all cases, synchronized cells rapidly
return to a normal cell cycle. Viability of synchronized hES cells relative to
unsynchronized cells when passaged and nucleofected was comparable when sub-cultured at
high density, while viability was reduced when sub-cultured at low density. Survival of
synchronized cells sub-cultured at low cell density required ROCK apoptosis inhibitor.
Importantly, the hES colonies that formed from the synchronized cultures had no apparent
changes in colony morphology; all colonies expressed high levels of alkaline
phosphatase, a marker for pluripotency.

Reviewer #2:

*The CRISPR/Cas9 system is rapidly revolutionizing the field of genome
engineering, allowing researchers to manipulate both coding and non-coding genomic
sequences at will in a constantly growing number of biological systems. This system
creates double-strand breaks (DSBs) at target loci, which can be repaired through one
of two cellular mechanisms: non-homologous end joining (NHEJ) or homology-directed
repair (HDR). The ability of a cell to repair a DSB generated by Cas9 through
HDR-mediated incorporation of exogenous DNA templates has recently been exploited to
engineer several modifications to endogenous loci, such as novel knock-in alleles,
point mutations and fluorescent tags, among others. However, the frequency of NHEJ is
usually higher than HDR due to the fact that NHEJ does not require any homologous or
exogenous DNA molecules to repair the DSB. Therefore, developing experimental methods
that increase the frequency of HDR is important in order for this technology to
fulfill its full potential in various laboratory studies as well as in clinical
applications*.

*Lin et al. have addressed whether cell cycle synchronization might affect the
relative use of these two repair pathways in an effort to define conditions that lead
to more efficient HDR. The authors tested whether reversible treatment of cells with
drugs reported to arrest cells in the S and late G2 cell cycle phases could increase
the rate of HDR when combined with timed delivery of Cas9-sgRNA ribonucleoprotein
complexes (RNPs) and various exogenous DNA templates. Using six pharmacological
agents and further narrowing the list down to two (nocodazole, which is reported to
block cells at late G2/M phase, and aphidicolin, which blocks cells at S phase), the
authors convincingly demonstrate that these treatments, coupled with timed delivery
of Cas9 RNPs and exogenous DNA templates, significantly increased the rate of HDR
across two loci in two different cell types. Importantly, they also convincingly
demonstrate that off-target editing is negligible using this approach*.

*This is a significant extension of this groups previous efforts aimed at
establishing the CRISPR/Cas9 genome editing system in mammalian cells for inducing
both NHEJ and HDR at specific genomic loci (*[13]*). The ability to
increase the rate of HDR through cell synchronization coupled with timed delivery of
Cas9 RNPs will undoubtedly have a significant impact in the field of genome editing,
particularly for applications aimed at engineering specific mutations of interest
into a variety of cell types, such as human ES and iPS cells*.

Major comments:

*1) The key observation here is that cells treated and released from different
chemical inhibitors of cell cycle progression undergo increased CRISPR/Cas9-mediated
HDR compared to untreated cells. While this is clearly shown, the cell cycle effects
that are associated with this treatment are not well characterized. Although the
drugs employed are commonly used in the field, it is important to characterize them
in the particular cell lines studied. The authors show cell cycle analysis in of
HEK293T cells in*
Figure 1—figure supplement 1*. However, for both nocodazole and aphidicolin treatment,
the data appear to show significant 2n as well as 4n peaks. Thus, it is unclear what
cell cycle phase might be associated with the increased HDR observed following
release from these treatments. Regarding the experiments with H9 human ES cells, in
which a combination of nocodazole and aphidicolin was used, there is no cell cycle
analysis shown at all. It will be important to address both of these issues prior to
publication*.

We now include a complete panel of cell cycle analysis for HEK293T, hES cells and the
newly added primary neonatal fibroblasts in Figure 1—figure supplement 1. We also performed alkaline phosphatase assay to
ensure the hES cells remain undifferentiated after synchronization. Regarding the
HEK293T nocodazole cell cycle block, we have redone the cell cycle analysis and find
that indeed the majority of cells are 4N. The same is true for primary fibroblasts and
hES cells. Some 4N HEK293T cells are slipping through the M-phase checkpoint and
initiating S-phase. This mitotic slippage phenomenon has been observed before in various
cell lines treated with nocodazole and other anti-microtubule drugs (Riffell et al.,
2009). No mitotic slippage was observed with nocodazole treatment of primary fibroblasts
or the hES cells that have intact cell cycle checkpoint regulation.

*2) With the exception of the experiments presented in*
Figure 2
*using the H9 cells, most of the experiments were carried out with HEK293T cells,
which are readily transfectable with nucleofection methods. Importantly, the rate of
HDR reported was significantly lower in drug-treated H9 cells compared to
drug-treated HEK293T cells. Moreover, induction of HDR in ES cells required a
modification of the protocol to incorporate a 16-hour pulse of nocodazole followed by
a 3-hour pulse of aphidicolin before Cas9 RNP nucleofection. One wonders how
generalizable these methods will be to other cell types. Therefore, the manuscript
would be strengthened with the addition of analysis of a panel of cell
lines*.

We agree that analysis of other cell lines is important to show that this method is
broadly applicable. We now include data for primary neonatal fibroblasts, a cell type
with low transfection efficiency. In this cell type, we observed enhanced total editing
and HDR with aphidicolin synchronization, in contrast to enhancement with nocodazole
treatment as observed in HEK293T and hES cells. Although these findings indicate some
variability according to cell type, the cell cycle synchronization procedure itself is
often not generalizable across different cell types. Due to variations in physiology,
growth rate and duration of cell cycle phases, one needs to determine and optimize the
synchronization protocol empirically. Nonetheless, the results presented here establish
the feasibility of timed delivery of Cas9 RNPs to enhance rates of site-specific genome
editing by homology-directed repair.

*3) The authors should establish the baseline nucleofection efficiencies for the
different cell lines tested. This will help clarify whether nucleofection efficiency
many be a contributing factor in the difference seen between HEK293T cells and H9
cells*.

Although nucleofection efficiency is likely to affect observed differences in
Cas9-mediated genome editing, we were not able to determine baseline RNP nucleofection
efficiencies for these cells.

*4) It is unclear whether the other cell cycle inhibitors besides nocodazole
shown in*
Figure 1—figure supplement 1
*were tested in ES cells. Minimally, this point should be clarified. If they were
not tested, is there a reason why not?*

We agree with the reviewer and have now clarified the manuscript by including a
statement about cell cycle synchronization in hES cells. In preliminary experiments, we
tested hES cells with the six cell cycle inhibitors. The results were disappointing,
with only nocodazole showing enhancement in total editing, but no HDR was detected.
Therefore, we adopted a method described by [21] in which hES cells were treated sequentially first with
nocodazole for 16h and then pulsed with aphidicolin for 3h, prior to nucleofection. With
this modification, higher levels of total editing and detectable HDR were observed.

*5) Given the interest in targeting efficiencies as a function of target loci, it
would be useful to extend this study to more than the two loci tested
here*.

We now include new data showing the editing efficiency in the CXCR4 gene in HEK293T
cells. Nocodazole synchronization led to markedly enhanced HDR efficiency. Similar to
EMX1 and DYRK1, the most significant increase was observed for cells receiving a lower
amount of Cas9 RNP. In this case, nocodazole synchronization yielded 27% HDR at 10 pmol
of Cas9 RNP. A comparable level of HDR in the unsynchronized cells would require 100
pmol of RNP. Enhancement of HDR at three different loci demonstrates that this timed
delivery of Cas9 RNP is a broadly applicable method in HEK293T cells.

*6) In*
Figure 3*, the authors
show that adding aphidicolin following release from a nocodazole block reduced HDR
efficiency in HEK293T cells, suggesting that S-phase entry may be required for
efficient HRD-mediated repair. They should show that this combined treatment actually
did block S-phase entry in these experiments, especially given the odd cell cycle
profiles shown in*
Figure 1—figure supplement 1. *Also, how does this conclusion jibe with the increased
efficiency of HDR in ES cells treated with this same combination when compared with
nocodazole alone?*

The experiments in Figure 3 third panel and hES
cells involved two different conditions. We thank the reviewer for pointing out the
confusion and we have now clarified this point in the manuscript.

In Figure 3, third panel, HEK293T cells were
synchronized with nocodazole prior to nucleofection. Immediately post nucleofection, one
dose of aphidicolin was added to the growth media to prevent the transfected cells from
proceeding into the S phase. The purpose was to reduce HDR efficiency, since the HDR
pathway is thought to be most active during S phase. We labeled this one-time addition
of aphidicolin “aphidicolin block” in Figure 3, as opposed to the standard aphidicolin synchronization procedure
used elsewhere in the manuscript. The standard aphidicolin synchronization procedure
involves treating the cells with the drug for 17h, releasing the cells for 7-8h, and
then treating the cells again for another 17h. Such sequential treatment does not fit
our experimental scheme, because cells were harvested 24h after nucleofection for
analysis. The goal of this experiment was to demonstrate that aphidicolin reduced the
HDR efficiency, instead of attempting to completely abolish the HDR. Our results in
Figure 3 second and third panels show that
the HDR frequencies were indeed significantly reduced.

The hES cells in Figure 3 were treated
differently. As described in the response above, we modified the standard one-drug
synchronization procedure in the HEK293T experiments to incorporate two drugs for
effective synchronization. We adopted a method described by [21] in which hES cells were treated
sequentially first with nocodazole for 16h and then pulsed with aphidicolin for 3h,
prior to nucleofection. After nucleofection, hES cells were grown in inhibitor-free
media.

*7) The authors argue in the Discussion that their approach of nucleofection of
Cas9 RNPs leads to higher cell viability than DNA transfection-based methods.
However, no data is shown to support this claim*.

We have now cited two published papers ([14], and Zuris et al., 2014), both of which have investigated the cell
viability between DNA- and RNP-based transfection methods.